# Deep Brain Stimulation of the Medial Septal Area Can Modulate Gene Expression in the Hippocampus of Rats under Urethane Anesthesia

**DOI:** 10.3390/ijms23116034

**Published:** 2022-05-27

**Authors:** Yulia S. Spivak, Anna A. Karan, Yulia V. Dobryakova, Tatiana M. Medvedeva, Vladimir A. Markevich, Alexey P. Bolshakov

**Affiliations:** Institute of Higher Nervous Activity and Neurophysiology, Russian Academy of Sciences, Moscow 117485, Russia; lampo_love@mail.ru (Y.S.S.); akartar.n@gmail.com (A.A.K.); julkadobr@gmail.com (Y.V.D.); golova93tanya@gmail.com (T.M.M.); v.markevich@yahoo.com (V.A.M.)

**Keywords:** deep brain stimulation, septum, dorsal hippocampus, ventral hippocampus, early genes, bdnf, ngf, inflammation

## Abstract

We studied the effects of stimulation of the medial septal area on the gene expression in the dorsal and ventral hippocampus. Rats under urethane anesthesia were implanted with a recording electrode in the right hippocampus and stimulating electrode in the dorsal medial septum (dMS) or medial septal nucleus (MSN). After one-hour-long deep brain stimulation, we collected ipsi- and contralateral dorsal and ventral hippocampi. Quantitative PCR showed that deep brain stimulation did not cause any changes in the intact contralateral dorsal and ventral hippocampi. A comparison of ipsi- and contralateral hippocampi in the control unstimulated animals showed that electrode implantation in the ipsilateral dorsal hippocampus led to a dramatic increase in the expression of immediate early genes (*c-fos, arc, egr1, npas4*), neurotrophins (*ngf, bdnf*) and inflammatory cytokines (*il1b* and *tnf*, but not *il6*) not only in the area close to implantation site but also in the ventral hippocampus. Moreover, the stimulation of MSN but not dMS further increased the expression of *c-fos, egr1, npas4, bdnf,* and *tnf* in the ipsilateral ventral but not dorsal hippocampus. Our data suggest that the activation of medial septal nucleus can change the gene expression in ventral hippocampal cells after their priming by other stimuli.

## 1. Introduction

The source of cholinergic innervation of the hippocampus, a brain area critically involved in memory formation, is the medial septal area (MSA), which consists of the medial septal nucleus and the diagonal band of Broca. The cholinergic neurons in this area are involved in the regulation of synaptic plasticity and rhythmogenesis in the neocortex and hippocampus [1,2]. Moreover, the activity of MSA neurons is important for the memory formation, learning, and spatial navigation [3,4,5,6,7,8,9,10]. Current views on the septohippocampal interaction, which are predominantly based on the electrophysiological analysis of neuronal activity in the MSA and hippocampus, suggest that acetylcholine released by MSA neurons induces short-term changes in the electrophysiological characteristics of hippocampal neurons and hippocampal synapses [11,12,13,14] and long-term changes in the efficacy of synaptic transmission in the hippocampus [1]. The ability of acetylcholine to induce long-term changes suggests that it can activate some signaling cascades that may finally result in changes in the expression of genes that are responsible for the development of long-term synaptic effects.

It was shown that the activation of muscarinic acetylcholine receptors induced transcription of the early gene *cyr61/ccn1* in HEK 293 cells [15,16]. The administration of M1 agonist pilocarpine, which is known to induce seizure activity, increased *c-fos* mRNA induction in many forebrain structures including piriform cortex, nucleus accumbens, amygdala, hippocampus, and neocortex [17]. It was shown that nicotinic acetylcholine receptors can also modulate gene expression in the hippocampal neurons via activation of the transcription factor CREB [18,19]. However, currently, it is largely unclear whether the activation of cholinergic neurons in MSA may lead to changes in the gene expression in the hippocampus. It was shown that the injection of potent glutamate agonist quisqualate into MSA leads to postponed elevation of mRNA expression of *bdnf* and *ngf* in the hippocampus [20]. Recently, it was shown that repeated electrical stimulation of MSA may induce the expression of *c-fos* in the dentate gyrus [21]. However, MSA is a heterogeneous structure, and it is known that the activation of different parts may result in different effects in the hippocampus [13,22], and the aforementioned studies did not analyze the consequences of activation of different parts of the MSA.

Deep brain stimulation (DBS) of the medial septal nucleus is considered as an approach that can have beneficial effects under pathological conditions [23]. In animal studies, deep brain stimulation of the medial septum improved spatial memory during cholinergic dysfunction [24]. Furthermore, it was shown that selective cholinergic activation triggers a robust network effect in the septo-hippocampal system during an inactive behavioral state [25]. One of possible long-term mechanisms that are triggered by deep brain stimulation of MSA may be related to the expression of immediate early genes, such as c-fos [21].

The aim of the present study was to examine the effect of deep brain stimulation at the different depths of the MSA, which correspond to the dorsal medial septum (dMS) and cholinergic medial septal nucleus (MSN), on the expression of genes in different parts of hippocampal formation.

## 2. Results

### 2.1. Electrical Stimulation of MSA at Different Depths Induces Different Field Response in the Hippocampus

Stimulation of the dorsal medial septum (dMS) at the depth of 3.5–4.5 mm evoked field responses in the CA1 region of the hippocampus. As depicted in Figure 1A, a negative waveform, having a latency of 3 msec, was recorded by a bipolar electrode positioned in the molecular layer of the hippocampus. After relocation of the stimulation electrode to the depth of 6.5 mm ventral to dura, this response disappears (Figure 1B). A position at this depth failed to elicit stable field responses. Nevertheless, in some animals, we observed positive/negative waves of excitation (Figure 1C). The amplitude of the responses evoked from this depth was lower compared with the amplitude at the higher coordinate.

### 2.2. Electrical Stimulation of MSA Does Not Affect EEG in the Hippocampus

To evaluate the possible general effect that may induce deep brain stimulation on the functioning of the hippocampus, we compared basic EEG characteristics in the hippocampus in the control and after dMS and MSN stimulation. We found that stimulation at both depths of the medial septum did not cause any significant effect on the rhythmic activity in the hippocampus (Figure 2).

### 2.3. Analysis of Gene Expression in the Hippocampus after Ultraslow Deep Brain Stimulation of MSA

In our experiments, we analyzed mRNA expression of several groups of genes that, according to the literature, may respond to elevation of acetylcholine level and, hence, to MSA stimulation. The first group consisted of immediate early genes such as *c-fos* [21], *arc, egr1, npas4,* and *cyr61* [15,16]. The second group included two major growth factors *bdnf* and *ngf* [20], and the third group included genes that encoded inflammatory cytokines *il1b, il6,* and *tnf* [26,27]. We compared the level of the mentioned mRNAs in the groups of animals, which were implanted with stimulation electrode in the dorsal medial septum (dMS) or MSN, and the control animals that were implanted with electrode in the MSN but were not stimulated. Our expression analysis included the four following structures: left dorsal and ventral hippocampi and right dorsal and ventral hippocampi. To analyze the effect of deep brain stimulation, we compared mRNA levels between experimental groups in each part of the hippocampus separately. In this comparison, we considered left hippocampi in the control unstimulated animals as the basal level of expression. We also compared mRNA expression between left and right parts of the hippocampus to elucidate additional effects that may arise from implantation of the recording electrode in the right dorsal hippocampus.

#### 2.3.1. Immediate Early Genes

We found that deep brain stimulation (DBS) of dMS or MSN did not induce any substantial influence on the expression of the studied immediate early genes in both parts of the left (intact) hippocampus (Figure 3). Electrode implantation in the control animals induced a strong increase in the expression of *c-fos* and *arc* in both dorsal and ventral parts of the right hippocampus compared to the corresponding parts of the left hippocampus. Importantly, DBS induced a significant increase in the right ventral hippocampus in the expression of *c-fos* and did not affect *arc* expression in both parts of the right hippocampus. The DBS-induced increase in *c-fos* expression occurred only after DBS of MSN but not dMS.

Electrode implantation also resulted in a significant increase in the expression of *egr1* and *npas4* in the right ventral hippocampus compared to the left ventral hippocampus. When we compared the dorsal parts of the left and right hippocampus in the control animals, we found the elevation of expression of these genes in the right dorsal hippocampus was not significant due to very high dispersion of mRNA expression between samples of the right dorsal hippocampus. However, post hoc analysis of the *egr1* expression in the right hemisphere showed that the increase after MSN stimulation but not dMS stimulation was significant for both genes (Figure 3). In the right ventral hippocampus, the effect of DBS was significant for both genes, and post hoc analysis revealed a significant increase in the expression of both *npas4* and *egr1* after DBS of MSN compared to control animals.

The expression of putative acetylcholine-dependent gene *cyr61/ccn1* in the right hippocampus was completely independent of DBS at any level of the medial septum; however, electrode implantation led to a significant decrease in the expression of this gene in the right ventral hippocampus compared to the left ventral hippocampus.

#### 2.3.2. Neurotrophins

Analysis of expression of *ngf* showed that the mRNA expression of this neurotrophin is not influenced by the DBS of any area of the medial septum. However, hippocampal damage resulting from electrode implantation led to an increase in the *ngf* expression in both right dorsal and ventral parts of hippocampus (Figure 4).

While *bdnf* expression in the undamaged left hippocampus was also not affected by DBS, an increase in the *bdnf* expression in the both parts of the right hippocampus caused by electrode implantation was significantly enhanced by DBS at the level of MSN but not at the level of dMS only in the ventral part, whereas the increase in the right dorsal hippocampus was insensitive to DBS (Figure 4).

#### 2.3.3. Inflammatory Cytokines

In our experiments, we analyzed changes in the expression of three cytokines involved in the regulation of inflammatory response *il1b*, *il6*, and *tnf* (Figure 5). The expression of all studied cytokines was not affected by DBS in the left undamaged hemisphere in both hippocampal parts. The expression of *il6* appeared to be insensitive to the damage of the right dorsal hippocampus by implanted electrode, and the subsequent stimulation of dMS and MSN did not influence its expression as well. In contrast, the expression of *il1b* was enhanced in the right dorsal hippocampus, which reflects the development of acute inflammation after electrode implantation into this hippocampal part. Stimulation of the medial septal area at both depths induced a trend to augmentation of expression of *il1b*; however, it did not reach the level of significance due to high dispersion among samples in both dMS and MSN groups. The picture of *tnf* expression in the right hippocampus was even more complex. It appeared that electrode implantation resulted in a significant increase in the *tnf* expression in the right ventral hippocampus and only a weak trend to enhancement in the right dorsal hippocampus despite the fact that implantation caused damage in the dorsal part. Stimulation of the septum at both levels induced a trend to an increase in the *tnf* expression in right dorsal hippocampus and did not affect the *tnf* level in the right ventral hippocampus.

## 3. Discussion

Septohippocampal interaction is frequently discussed in terms of the generation of hippocampal rhythms and plasticity of hippocampal synapses, which are critical for learning and memory formation [28]. Previous studies showed that memory formation is associated with changes in the expression of immediate early genes, suggesting that the activation of medial septal neurons may be one of factors that induces changes in the gene expression in the hippocampal cells. In our study, we used the stimulation of two parts of the medial septum to evaluate its effect on the functioning of the hippocampus. We found, in agreement with previous reports [22], that the stimulation of dMS induces field responses in the hippocampus, and electrode implantation in the medial septal nucleus practically does not induce any field response in the hippocampus, which is probably reflecting the fact that the electrode reached the area that predominantly contains cholinergic neurons [13,29]. Stimulation of both parts of the medial septum did not induce any effect on the EEG pattern in the hippocampus, supporting previous data that medial septum stimulation has a very narrow temporal window for influencing characteristics of hippocampal neurons [13].

We found that the stimulation of dMS and MSN had no effect on the expression of all studied genes in the left dorsal and ventral hippocampi, which were not damaged by the electrode implantation. In contrast, in the right hippocampus, which was damaged by the electrode implantation, we found that the stimulation of MSN but not dMS modulated the expression of several studied genes. First, it should be mentioned that stimulation of the medial septum induces acetylcholine release in both the left and right hippocampi [30], and the observed asymmetry of response is not related to any asymmetry of acetylcholine release. Second, the electrode implantation per se induced not only local effect in the dorsal hippocampus, where it was implanted, but also a distant effect in the ventral hippocampus. At first glance, electrode implantation should have a local effect in the dorsal hippocampus where it should induce an inflammatory response. Indeed, we observed an increase in the expression of proinflammatory cytokines typical of acute inflammatory process in the dorsal hippocampus (*il1b* and *tnf*). However, the increase in the expression of *il1b* was localized to the dorsal hippocampus, whereas *tnf* expression increased in the ventral part of the right hippocampus, suggesting that an additional process occurred that spreads the effect from the dorsal to ventral part of the hippocampus without spreading to the contralateral hippocampus. It is well known that the hippocampal tissue is very sensitive to the induction of spreading depression, and its injury may induce spreading depression [31]. After induction, spreading depression will travel along the longitudinal axis of the hippocampus and will not be induced in the contralateral hemisphere. It was previously shown that in the neocortex, spreading depression increases the expression of *c-fos* [32], *arc* [33], *egr1* [34], *npas4* [35], *bdnf* [33], *il1b* [36], and *tnf* [36]. Our data are in agreement with these observations because in our case, the expression of all these genes, except *il1b*, increased in the right ventral hippocampus compared to the left ventral hippocampus. Additionally, we showed that the expression of early gene *cyr61/ccn1* in the ventral hippocampus decreases after putative spreading depression. Presumably, spreading depression resulted from tissue damage after electrode implantation in the right dorsal hippocampus and spread to the ventral hippocampus, where it altered the expression of the majority of studied genes.

As we mentioned above, stimulation of the medial septal area at different depths induces different field responses in the hippocampus. Stimulation in the dorsal medial septum induced a clear field excitatory postsynaptic response in CA1 area; however, it had no effect on the expression of any of the studied genes in any of the studied hippocampal areas. In contrast, stimulation of the medial septal area at a depth of 6.5 mm induced a barely visible field response; however, it modulated the expression of a majority of the studied genes in the ventral hippocampus. This deep septal area is the location of the medial septal nucleus rich in the cholinergic neurons. Presumably, stimulation of this area led to activation of the septal network and release of acetylcholine in the hippocampus.

One of our unexpected findings is that deep brain stimulation by trains of paired pulses can induce changes in the expression of genes only in the hippocampus that was potentially affected by spreading depression after electrode implantation. The latter means that the sensitivity of hippocampal cells was shifted by spreading depression, and even weak stimulus coming from the MSN was able to enhance the expression of immediate early genes *c-fos*, *egr1*, and *npas4* but not *arc* and *cyr61*. The expression of inflammatory cytokines was largely left unchanged, except for the expression of *tnf* in the right dorsal hippocampus, where we observed only a trend to an increase after MSN stimulation. The effect of septal stimulation on the expression of *bdnf* and *ngf* also was not the same. While *ngf* expression remained insensitive to MSN and dMS stimulation in both hippocampi, the *bdnf* expression increased specifically in the right ventral hippocampus after MSN but not dMS stimulation. Taken together, these data suggest that MSN stimulation has a specific modulatory effect on not all transcriptional machinery but is likely to specifically activate the transcription of some genes in cells located in specific part of the hippocampus. The mechanism of sensibilization of hippocampal cells to the putative acetylcholine release from septal fibers after the passing of spreading depression in the hippocampus may include epigenetic changes that occurred after a massive release of various transmitters (glutamate, dopamine, NO, etc. [37,38,39,40]) and an increase in the density of muscarinic acetylcholine receptors after spreading depression [41]. The selective sensitivity of the ventral hippocampus to MSN DBS, compared to the dorsal part, is unclear but may be related to a well-known difference in the expression of various proteins between these hippocampal parts [42,43].

An important outcome of our results is that a train of paired stimuli activating MSN can alter the expression of a number of genes in the hippocampal cells that were previously sensitized by another stimulus (in our case, spreading depression). These results suggest that the induction of long-term effects in the hippocampal cells after MSN activity requires the specific activation of hippocampal cells before the arrival of MSN input. If the activation of hippocampal cells is strong enough to alter the epigenetic state of cells, then several trains of paired activation of MSN, which previously had no effect, will be able to change gene expression. From a physiological viewpoint, our DBS protocol seems to be very weak because septal neurons have predominantly bursting activity at frequencies above 4 Hz [44]; however, even this activity can produce a long-term effect in the cells that were “made ready” for receiving this stimulus. It is possible to hypothesize that under physiological conditions, a situation with the induction of changes in the gene expression by the activity of septal neurons will be similar, i.e., the activity of hippocampal cells in combination with inputs to these cells from other brain parts can sensibilize these cells to MSN activity, which can lead to changes in the expression of genes.

From a functional viewpoint, the early genes *c-fos* and *egr1* are transcriptional regulators, whose expression can trigger intracellular cascades leading to the expression of other genes, including *arc* and *bdnf* [45], which, in turn, are important regulators of synaptic plasticity [45,46]. *Npas4* is also a transcription factor that is activated by neuronal depolarization and modulates the expression of genes responsible for the regulation of the excitation/inhibition balance, including *bdnf* [47]. The enhancement of expression of these early transcription factors after MSN stimulation suggests that the activity of septal neurons can intensify the response of hippocampal cells to other stimuli and trigger or prolong plastic processes in these cells under normal conditions by the augmentation of expression of early genes involved in nerve cell plasticity. Stress is one of the well-known inducers of plastic changes in the CNS. Previously, it was shown that acute stress may induce the expression of early genes including *egr1* [46] and *c-fos* [48], and it was proposed that the activation of septal cholinergic neurons may be one of inductors of expression of *c-fos* [48]. Here, we show that the activation of the medial septal input to the hippocampus can, at least, enhance the expression of these early genes, pointing to the possibility that cholinergic activity indeed can be one of factors that can upregulate the expression of early genes in the hippocampus under stress conditions.

Among the genes we studied, *cyr61* and *ngf* were the most probable candidates for being acetylcholine-dependent because it was previously shown that *cyr61/ccn1* is an early gene whose expression may be activated in culture by acetylcholine [49], and ngf expression may be induced in the hippocampus by the activation of MSA [20]. We found that the level of mRNA of these genes is insensitive to the stimulation of either dMS or MSN. We believe that at higher stimulation frequencies, the enhancing effect of MSN activity may appear.

It is also important to note that there are some data that acetylcholine may serve as a regulator of inflammatory processes in the nervous tissue by influencing the function of glial cells [26,27]. However, our data suggest that the low-frequency activity of septal neurons can hardly affect the expression of inflammatory cytokines during acute inflammation. There may be several reasons for the absence of this effect. First, according to single-cell RNAseq data [50,51,52,53], in the mouse brain, all types of acetylcholine receptors are predominantly expressed in neurons, whereas the studied pro-inflammatory cytokines are expressed in glial and vascular cells. However, the modulatory effect of acetylcholine was mainly described in cultured glial cells, which may differ from the brain cells in vivo by the expression profile of acetylcholine receptors. Second, a more substantial increase in the level of acetylcholine may be required to produce noticeable changes in the level of inflammatory cytokines. Third, the protective effect of acetylcholine may appear under chronic inflammatory conditions when the sensitivity of glial cells to this mediator is changed. Anyway, under conditions of acute inflammation, slow septal activity can hardly be used to modulate inflammation in the brain.

## 4. Materials and Methods

The experiments were performed with adult male Wistar rats (250–350 g) received from the Research Center of Biomedical Technology RAMS, nursery “Pushchino.” A total of 24 rats were involved in the study (*n* = 8/group). Animals were housed under standard vivarium conditions at 21 ± 1 °C with a 12 h light/dark cycle; food and water were provided ad libitum. All experiments were performed in accordance with the ethical principles stated in the EU Directive 2010/63/EU for animal experiments and were approved by the Ethical Committee of the Institute of Higher Nervous Activity and Neurophysiology of the Russian Academy of Sciences.

### 4.1. Stereotaxic Surgery and Electrophysiology

Rats were anesthetized with urethane (1.75 g/kg, i.p.) and mounted in a Kopf stereotaxic frame for surgical preparation for the recording session. A stimulating nickel–chrome electrode (diameter 80 μm) was implanted into the medial septal area (0.5 mm posterior, 0.0 mm lateral to bregma, approximately 3.5–4.5 mm or 6.5 mm ventral to dura). A recording electrode was placed into the CA1 area (2.7 mm posterior, 1.5 lateral to bregma, 2.2 mm ventral to dura) [54]. One electrode under the skin served as a ground and as a reference electrode.

The fEPSP amplitude in the CA1 field evoked by paired MS stimulation (interstimulus interval 30 ms; intertrain time 20 s at intensity of 100–300 μA; 10 paired stimulations) was recorded every 10 min for 1 h. The intensity of the testing paired pulse stimulation was set to evoke 40–50% of the maximum fEPSP amplitude. In our experiments, for long-term recordings, we applied urethane anesthesia, which is used for non-recovery procedures of exceptionally long duration where the preservation of autonomic reflexes is essential and thus does not need any additional euthanasia procedure.

### 4.2. EEG Recording

EEGs were recorded using the same electrode that was placed into the CA1 area (see Stereotaxic surgery and electrophysiology), with low-pass and high-pass filters of 1 kHz and 5 Hz, respectively. Seven sessions of 80 s took place during the experimental procedure. Each 80 s session was conducted after recording of the fEPSP amplitude in the CA1 field evoked by paired MS stimulation.

### 4.3. RNA Isolation and Reverse Transcription

Tissue samples were collected in 1 h after the start of stimulation, placed in 1.5 mL tubes, and frozen in liquid nitrogen. RNA isolation was performed using an ExtractRNA reagent (Evrogen, Moscow, Russia) in accordance with the manufacturer’s recommendations. To remove traces of genomic DNA, RNA samples were treated with DNase I (Thermo Scientific, Vilnius, Lithuania). Reverse transcription was performed using the MMLV RT reagent kit (Evrogen, Moscow, Russia) and murine RNase Inhibitor (New England Biolabs, Ipswich, MA, USA) as recommended by the manufacturer. An equimolar mixture of random decaprimer (Evrogen, Moscow, Russia) and oligo(dT)15 primer (Evrogen, Moscow, Russia) was used; the concentration of each primer in the reaction was 1 μM. After reverse transcription, the reaction mixture was diluted 8-fold with deionized water.

### 4.4. qPCR

Relative quantities of mRNAs for the genes of interest were evaluated with a BioRad CFX384 real-time PCR station (BioRad, Singapore) using a qPCRmixHS SYBR + LowROX mix for PCR (Evrogen, Moscow, Russia) according to the manufacturer’s recommendations. Relative quantities of mRNAs were normalized to the to the geometric mean of the mRNA expression levels for the *ywhaz* and *osbp* genes. The quality of the DNase treatment was evaluated in all the samples and genes by performing a negative control qPCR with the product of DNase I treatment. Primers for qPCR were designed for the mRNA sequences (Table 1) from the NCBI database using the PrimerSelect software package (DNASTAR Lasergene). Gene expression was analyzed by the E^−ΔΔCt^ method.

### 4.5. Data Analysis

The data are presented as mean  ±  SD. All groups were examined for the presence of outliers using the interquartile range method; the outlying values were automatically excluded from analysis by software. Since data distribution did not pass the normality Shapiro–Wilk test, the significance of differences between groups was evaluated using the Kruskal–Wallis ANOVA (the level of significance was *p*  <  0.017 for comparison of three groups) followed by Dunn’s post hoc test (the level of significance was *p*  <  0.025) using scripts on R with commands kruskal.test() и dunn.test(). In the case of comparison of two groups, the Mann–Whitney test was used (the level of significance was *p*  <  0.05).

## Figures and Tables

**Figure 1 ijms-23-06034-f001:**
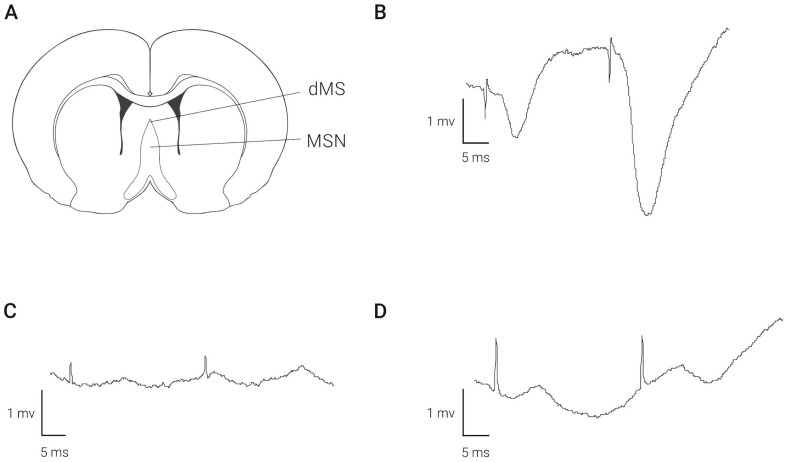
Field responses recorded in the hippocampal CA1 area following stimulation of medial septal area at different depths. Panel (**A**) shows localization of stimulating electrodes. Examples of fEPSP after stimulation of dMS (**B**) and MSN (**C**,**D**) are shown.

**Figure 2 ijms-23-06034-f002:**
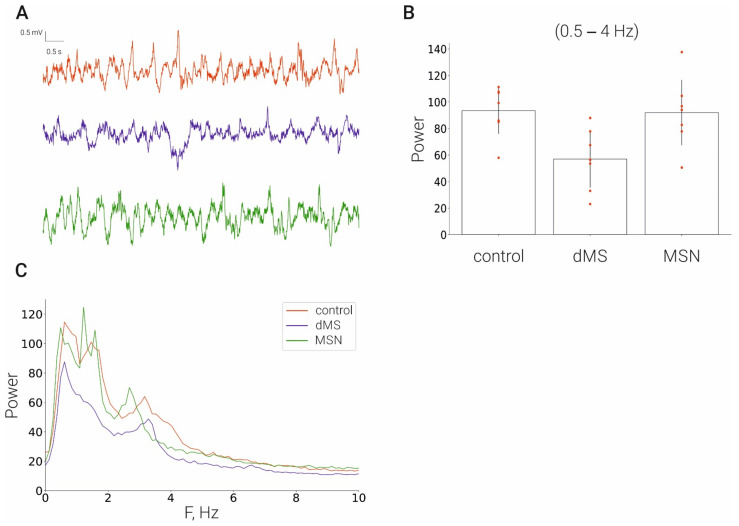
Effect of stimulation of the medial septum on the hippocampal EEG. (**A**), LFP time series of: control rats (red line), dMS rats (blue), MSN rats (green); (**B**), corresponding power spectra; (**C**), corresponding averaged power in frequency range [0.5–4] Hz.

**Figure 3 ijms-23-06034-f003:**
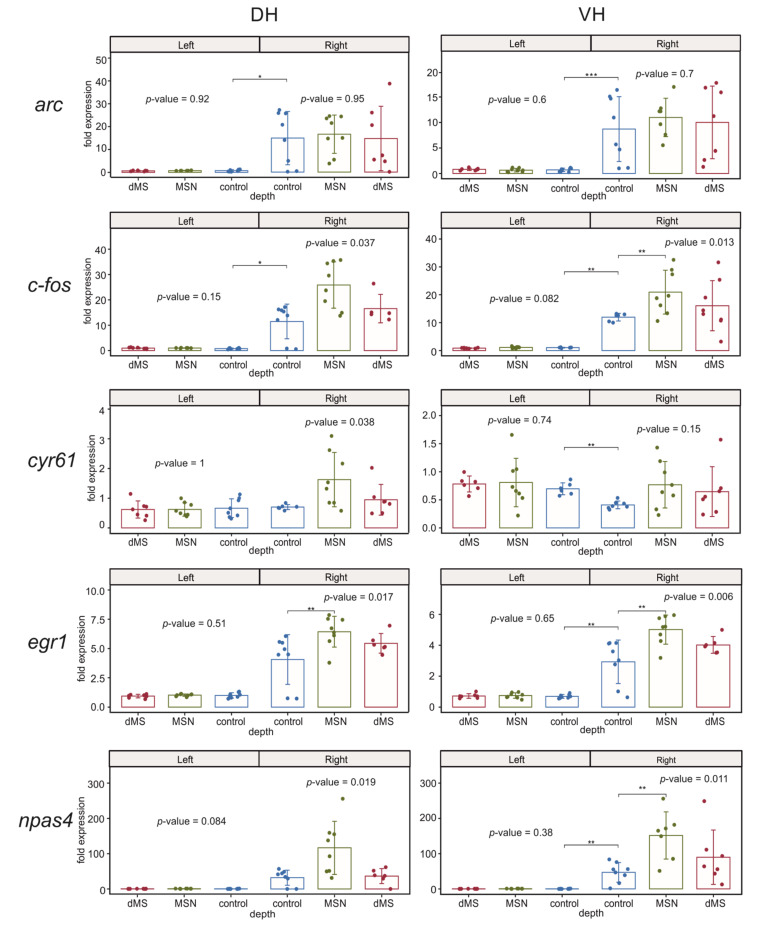
Changes in the expression of early genes after stimulation of the dorsal medium septum (dMS) and medial septal nucleus (MSN) in dorsal and ventral parts (DH and VH, respectively) of the left (intact) and right (damaged by implanted electrode) hippocampi. *p*-values are shown for Kruskal–Wallis test for comparison of three groups in one hemisphere. Interhemispheric comparisons were performed only in the control group using the Mann–Whitney test. For the intergroup comparisons, *, **, and *** mark significant differences at 0.005 ≤ *p* < 0.025, *p* < 0.005, and *p* < 0.0005 (post hoc Dunn test). For interhemispheric comparisons, *, **, and *** mark significant differences at *p* ≤ 0.05, *p* < 0.01, and *p* < 0.001 (Mann–Whitney test).

**Figure 4 ijms-23-06034-f004:**
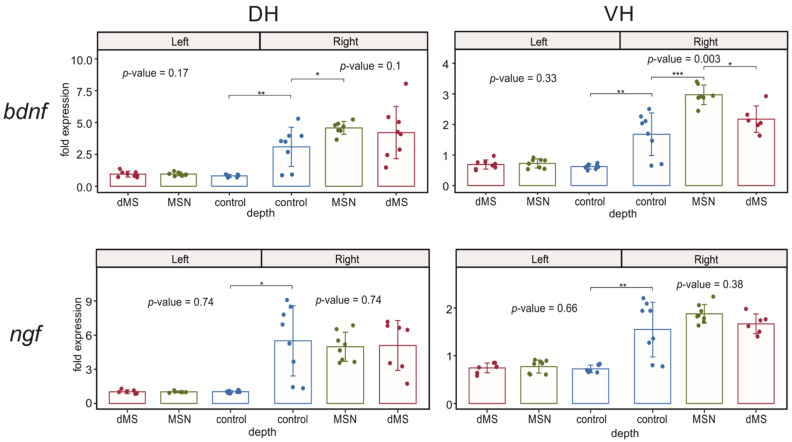
Changes in the expression of neurotrophins after stimulation of the dorsal medium septum and medial septal nucleus in dorsal and ventral parts (DH and VH, respectively) of the left (intact) and right (damaged by implanted electrode) hippocampi. *p*-values are shown for Kruskal–Wallis test for comparison of three groups in one hemisphere. Interhemispheric comparisons were performed only in the control group using the Mann–Whitney test. For the intergroup comparisons, *, **, and *** mark significant differences at 0.005 ≤ *p* < 0.025, *p* < 0.005, and *p* < 0.0005 (post hoc Dunn test). For interhemispheric comparisons, *, **, and *** mark significant differences at *p* ≤ 0.05, *p* < 0.01, and *p* < 0.001 (Mann–Whitney test).

**Figure 5 ijms-23-06034-f005:**
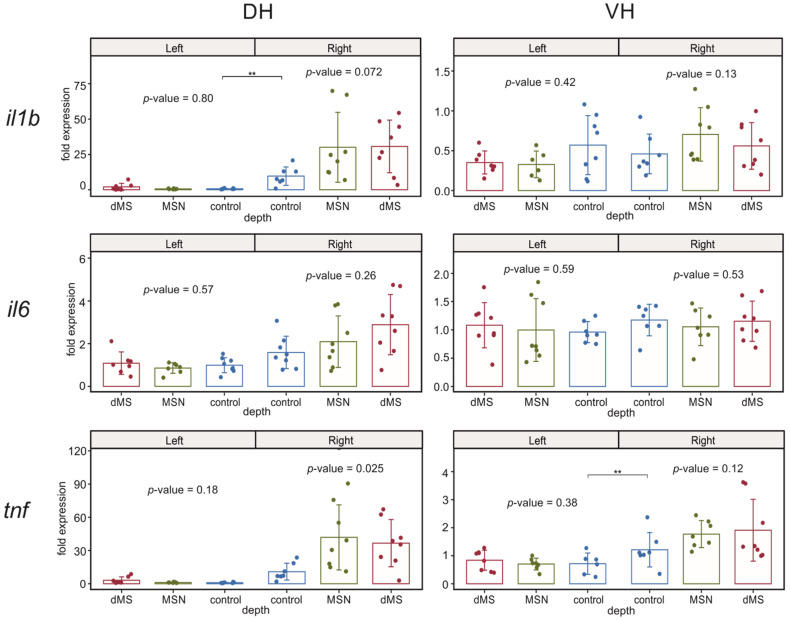
Changes in the expression of inflammatory cytokines after stimulation of the dorsal medium septum and medial septal nucleus in dorsal and ventral parts (DH and VH, respectively) of the left (intact) and right (damaged by implanted electrode) hippocampi. *p*-values are shown for Kruskal–Wallis test for comparison of three groups in one hemisphere. Interhemispheric comparisons were performed only in the control group using the Mann–Whitney test. For interhemispheric comparisons, ** mark significant differences at *p* < 0.01 (Mann–Whitney test).

**Table 1 ijms-23-06034-t001:** Primer sequences for qPCR.

Gene	Forward Primer	Reverse Primer	T_annealing_, °C
*Osbp*	TCC GGG AGA CTT TAC CTT CAC TT	GTG TCA CCC TCT TAT CAA CCA CC	65
*Ywhaz*	TTG AGC AGA CGG AAG GT	GAA GCA TTG GGG ATC AAG AA	63
*Fos*	CAAAGTAGAGCAGCTATCTCC	CTCGTCTTCAAGTTGATCTGT	63
*Arc*	GCAGGTGGGTGGCTCTGAAGAATA	TCCCGCTTACGCCAGAGGAACT	69
*Egr1*	ACC CAC ATC CGC ACC CAC ACA	GCA GCT GAG GCC ACG ACA CT	62.5
*Npas4*	GTG GAC GTC CCC CTG GTG CC	CCT GTC CAT GCC CTG AGC CAA C	62.5
*Cyr61/ccn1*	CAG CCC TGC GAC CAC ACC AAG	CAG CCC ACA GCA CCG TCA ATA CA	62
*Ngf*	AGC ACC CAG CCT CCA CCC ACC TC	CTC GCC CAG CAC TGT CAC CTC CTT	66.5
*Bdnf*	CCA TAA GGA CGC GGA CTT GTA C	AGA CAT GTT TGC GGC ATC CAG G	63
*Il1b*	TCT GTG ACT CGT GGG ATG AT	CAC TTG GCT TAT GTT CTG TC	61
*Il6*	GCC ACT GCC TTC CCT ACT TCA C	GAC AGT GCA TCA TCG CTG TTC ATA C	63
*Tnf*	GTC CAA CTC CGG GCT CAG AAT	ACT CCC CCG ATC CAC TCA G	65

## Data Availability

Raw data are available from the authors upon request.

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
