# Peer review of "Deep Brain Stimulation of the Medial Septal Area Can Modulate Gene Expression in the Hippocampus of Rats under Urethane Anesthesia"

_ijms, 2022, doi:10.3390/ijms23116034_

Round 1

Reviewer 1 Report

Journal: IJMS-1724025

Title: Deep brain stimulation of the medial septal area can modulate gene expression in the hippocampus of rats under urethane anesthesia

The investigation analyzes the effects of deep brain stimulation of the medial septal area on the gene expression in rats' dorsal and ventral hippocampus. Animals under urethane anesthesia were implanted with a recording electrode in the right hippocampus, a stimulating electrode in the dorsal medial septum (dMS), or medial septal nucleus (MSN). Ipsilateral contralateral dorsal and ventral hippocampi were collected following one-hour-long deep brain stimulation.

Using quantitative PCR, control unstimulated animals showed that electrode implantation in the ipsilateral dorsal hippocampus led to a dramatic increase in the expression of immediate early genes (c-fos, arc, egr1, npas4), neurotrophins (ngf, bdnf) and inflammatory cytokines (il1b and tnf, but not il6) not only in the area close to implantation site but also in the ventral hippocampus. However, deep brain stimulation did not cause any changes in the intact contralateral dorsal and ventral hippocampi. Interestingly, stimulation of MSN but not dMS further increased expression of c-fos, egr1, npas4, bdnf, and tnf in the ipsilateral ventral but not dorsal hippocampus, suggesting that the MSN activation can increase the gene expression in ventral hippocampal cells.

The research objectives and results are presented and discussed adequately, suggesting that deep brain stimulation can induce the expression of specific genes in precise areas of the hippocampus. There are a few shortcomings the authors have to address.

  1. ACRONYMS: Line 55: medial septal nucleus (MSN). Line 287: medial septal area (MS). Line 87-88: medial septal area. DBS (Deep brain stimulation?). Please review the concordance of the acronyms throughout the text. The difference between MSN and MSA in the main text is not clear.
  2. Figure 2 shows the average power is lower in dMS group at low frequencies. Do the authors have any explanation?
  3. Line 93: "In our experiments, we analyzed mRNA expression of several groups of genes that, according to literature, may respond to MSA stimulation." Could the authors add some essential references to the issue?
  4. Also, it would be interesting to include a brief paragraph regarding the principal roles of the genes investigated. In addition to the mRNA expression of the investigated genes, do the authors have in mind to quantify the corresponding proteins or their immunohistochemical expression?
  5. Line 160: "…the expression of three popular cytokines…" The word "popular" does not fit very well in this context. Please change “popular” by "prevalent" or something like that.
  6. Line 188: "Previous studies showed that memory formation is associated with changes in the expression of immediate early genes…" Please add some appropriate references to the issue.
  7. Regarding the mechanism that mediates the effect of DBS on gene expression, it seems that spreading depression and acetylcholine release may play a critical role in the acute experimental setting (1 hour after the start of stimulation) presented in the manuscript. However, in chronic DBS, many different functional gen categories have been demonstrated, including some involved in ion channel activity, gated channel activity, calcium signaling pathway, and glutamatergic synapse closely related to neuronal activity. Do the authors know the up-expression or down-expression of specific genes associated with the astroglial function in the neocortex or the hippocampus?
  8. The authors argue the selective sensitivity of the ventral hippocampus (compared to the dorsal part) to MSN deep brain stimulation may be related to differences in the architectural configuration and, therefore, in the expression of different proteins. What role can neuroglia gene expression play here? Are there studies that address neuronal and astroglial differential expression secondary to DBS?
  9. A final paragraph is needed to explain the potential consequences of the reported findings in human physiological and pathological conditions.

Reviewer 2 Report

A very interesting and novelty work. I have as a minor suggestion, that the authors should discuss the limitations of their work. 

Author Response

We are grateful to referee for critical reading of our manuscript.